# Response of Photosynthetic Capacity to Nitrogen Addition in *Larix gmelinii* Trees in Different Crown Classes

**DOI:** 10.3390/plants14071056

**Published:** 2025-03-28

**Authors:** Ruijia Cai, Jingjing Wang, Rui Zhang, Qinggui Wang, Chuankuan Wang, Xiankui Quan

**Affiliations:** 1Key Laboratory of Sustainable Forest Ecosystem Management, Ministry of Education, Northeast Forestry University, 26 Hexing Road, Harbin 150040, China; crj1216@163.com (R.C.); a15803474487@163.com (J.W.); zhangr0120@163.com (R.Z.); wangck-cf@nefu.edu.cn (C.W.); 2School of Life Sciences, Qufu Normal University, 57 Jingxuan West Road, Qufu 273165, China; wangqinggui@hlju.edu.cn

**Keywords:** photosynthesis, nitrogen deposition, crown class, boreal forest

## Abstract

We explored the response of photosynthetic capacity to nitrogen (N) deposition among *Larix gmelinii* trees in different crown classes (e.g., suppressed, intermediate, and dominant trees) in a 12-year field experiment in a forest in the Greater Khingan Mountains in Northeast China. Four N-addition treatments were established: control (CK), low N (LN), medium N (MN), and high N (HN) (0, 25, 50, and 75 kg N·ha^−1^·year^−1^, respectively). Photosynthesis and its influencing factors were measured in 2023. Nitrogen addition significantly increased the maximum net photosynthetic rate (*P*_max_), maximum carboxylation rate (*V*_cmax_), and maximum electron transport rate (*J*_max_) of suppressed and intermediate trees. The suppressed trees showed maximum *P*_max_ and *V*_cmax_ in MN and HN, and maximum *J*_max_ in HN. The intermediate trees showed maximum *P*_max_, *V*_cmax_, and *J*_max_ in MN. For dominant trees, *P*_max_ was increased in LN and MN and decreased in HN, and *V*_cmax_ was increased by N addition and peaked in MN. Nitrogen addition significantly increased the leaf N content (N_mass_), chlorophyll content (Chl_m_), the ratio of N to phosphorous (N:P), and photosynthetic enzyme activities in all crown classes. N_mass_ had significant nonlinear relationships with *P*_max_, *V*_cmax_, and *J*_max_. Enzyme activity and Chl_m_ positively affected the photosynthetic capacity of suppressed and intermediate trees, and N:P negatively affected the photosynthetic capacity of dominant trees. The promoting effect of N addition on photosynthetic capacity was stronger in suppressed and intermediate trees than in dominant trees. Therefore, the crown class should be considered when studying the effect of N deposition on the boreal forests.

## 1. Introduction

Photosynthesis is the most fundamental process of material production and energy metabolism in trees and is typically positively associated with the leaf nitrogen (N) content [1,2]. Anthropogenic N deposition has increased N availability in forests, potentially influencing the photosynthetic capacity of trees [3,4,5]. A number of studies have examined the effect of N addition on the photosynthetic capacity of trees and have obtained diverse results, ranging from a promoting effect [6,7,8] to an inhibiting effect [9]. This may be because when the N supply exceeds the biological N demand, trees change from an N-limited to an N-saturated state. This change in N status can affect their photosynthetic capacity [10,11]. Some studies have shown that key photosynthetic enzymes are affected by N supply, and there is a critical point in the response of photosynthesis to N addition [12,13]. Prior N-addition studies have mainly focused on standard trees and ignored differences among trees of the same species [14,15,16]. Especially in natural forests, trees of the same species are present in a range of crown classes because of the influence of succession and competition. Dominant trees are more competitive than suppressed trees for light, water, and other resources. The addition of N may alleviate the resource competition pressure on suppressed trees, thereby changing forest structure. The results of studies on the effects of N addition on the photosynthetic capacity of trees in different crown classes in natural forests are inconclusive. Exploring the responses of trees in different crown classes to N addition can not only provide ideas for forest management, but also reveal the impact of N deposition on forest structure.

Boreal forests account for about 30% of the global forest area [17], and are among the ecosystems that are seriously N-limited [18]. Dahurian larch (*Larix gmelinii*) is a widely distributed and dominant species in Eurasian boreal forests. At the time of publication of the China Forest Resources Report (2014–2018), the area and stock volume of natural *L*. *gmelinii* forest was 7.08 million ha and 766 million m^3^, accounting for 5.76% and 5.61% of the total in China, respectively [19]. Therefore, the effects of N enrichment on the photosynthesis of *L*. *gmelinii* need to be evaluated.

To date, the response of *L*. *gmelinii*’s photosynthetic capacity to N addition has mainly been studied in plantations and seedlings [20,21]. Although a few studies have focused on natural forests, none has discriminated among trees in different crown classes [22,23]. Research on the effects of N addition on the photosynthetic capacity of trees in different crown classes is still lacking. In 2011, according to the atmospheric N deposition rate in northern China (25 kg N·ha^−1^·year^−1^) [24], we established four N addition treatments (0, 25, 50, 75 kg N·ha^−1^·year^−1^) to simulate increases in atmospheric N deposition by 0, 1, 2, and 3 times. Twelve years later, in 2023, we examined the responses of photosynthetic capacity to N addition and explored the variations in responses among trees in different crown classes. We tested the following hypotheses: (1) N addition will enhance photosynthetic capacity by increasing leaf N content and enzyme activity; (2) The effects of N addition on photosynthetic capacity will vary significantly among crown classes, and suppressed trees will respond positively to N addition.

## 2. Results

### 2.1. Comparison of Photosynthetic Characteristics

Nitrogen addition significantly (*p* < 0.05) affected the *P*_max_, *V*_cmax_, *J*_max_, and *PNUE* (Table 1), and the interaction between N addition and crown class was significant for all these indexes. The *P*_max_, *V*_cmax_, *J*_max_, and *PNUE* of suppressed and intermediate trees were significantly higher (*p* < 0.05) in the LN, MN, and HN treatments than those in the control treatment, with the exception of *J*_max_ of intermediate trees in the LN treatment (Figure 1, Table 1). The suppressed trees had the maximum *P*_max_, *V*_cmax_, and *PNUE* in MN and HN treatments, and had the maximum *J*_max_ in HN treatment (Figure 1). For dominant trees, the maximum *P*_max_ and *J*_max_ were in the MN treatment, and the maximum was in the HN treatment (Figure 1a,c). The *V*_cmax_ of intermediate trees showed no significant difference between LN, MN, and HN treatments (Figure 1b). For dominant trees, the *P*_max_ and *PNUE* were significantly higher (*p* < 0.05) in the LN and MN treatments and significantly lower in (*p* < 0.05) in the HN treatment, compared with CK (Figure 1a,d). Nitrogen addition significantly (*p* < 0.05) increased the *V*_cmax_ of dominant trees, and the maximum value was in the MN treatment (Figure 1b). However, N addition did not affect the *J*_max_ of dominant trees (Figure 1c). The maximum values of *P*_max_ for suppressed, intermediate, and dominant trees were 4.63 μmol·m^−2^·s^−1^ (HN), 8.79 μmol·m^−2^·s^−1^ (MN), and 6.44 μmol·m^−2^·s^−1^ (MN), respectively (Figure 1a).

In CK, the *P*_max_, *V*_cmax_, *J*_max_, and *PNUE* of suppressed trees were significantly (*p* < 0.05) lower than those of intermediate and dominant trees (Figure 1). The differences in *P*_max_, *V*_cmax_, *J*_max_, and *PNUE* between suppressed trees and dominant trees gradually decreased as the amount of added N increased. Especially in the HN treatment, the *P*_max_, *V*_cmax_, *J*_max_, and *PNUE* of suppressed trees were significantly (*p* < 0.05) higher than those of dominant trees (Figure 1).

In LN treatment, the increasing rates in *J*_max_ and *V*_cmax_ of suppressed trees were higher than those of intermediate and dominant trees, and the increasing rate in *P*_max_ did not show significant difference among crown classes (Figure 2). In MN and HN treatments, suppressed trees had higher increasing rates in *J*_max_, *V*_cmax_, and *PNUE* than intermediate and dominant trees, and suppressed and intermediate trees had higher increasing rate in *P*_max_ than dominant trees. The HN treatment had an inhibitory effect on the *P*_max_ and *PNUE* of dominant trees (Figure 1a,d). The *P*_max_ of suppressed trees was increased by 26.09%, 52.31% and 55.73% in the LN, MN, and HN treatments, respectively, compared with CK; and the *P*_max_ of intermediate trees was increased by 20.98%, 72.35% and 54.71% in LN, MN, and HN treatments, respectively, compared with CK (Figure 2). The *P*_max_ of dominant trees was increased by 23.89% and 41.26% in the LN and MN treatments, respectively, and decreased by 11.83% in the HN treatment, compared with CK (Figure 2).

### 2.2. Comparison of Factors Related to Photosynthetic Capacity

The interaction between N addition and crown class was significant for Chl_m_, N_mass_, and N:P (*p* < 0.01, Table 1). Compared with CK, the LN, MN, and HN treatments significantly (*p* < 0.01) increased the Chl_m_, N_mass_, and N:P for all crown classes. As the amount of added N increased, the N_mass_ and N:P increased for all crown classes, and the Chl_m_ exhibited a trend of first rising and then falling for intermediate and dominant trees (Figure 3a,b,d). Nitrogen addition did not significantly affect P_mass_ in all the crown classes (Figure 3c).

In CK, the suppressed trees had lower N_mass_ and N:P than the intermediate and dominant trees, and the intermediate trees had higher Chl_m_ than the suppressed and dominant trees (Figure 3a,b,d). In the LN, MN, and HN treatments, dominant trees had higher N_mass_ and N:P than suppressed and intermediate trees (Figure 3b,d), and the suppressed trees had higher Chl_m_ than dominant trees (Figure 3a). The intermediate trees had higher Chl_m_ than dominant trees in the LN treatment and had lower Chl_m_ than suppressed trees in the HN treatment (Figure 3a). There were no significant differences in P_mass_ among crown classes in each N addition treatment (Figure 3c).

Nitrogen addition significantly affected the activities of RubisCO, GS, GLO, and PEPC (*p* < 0.01, Table 1). For suppressed trees, the activities of RubisCO, GS, and GLO increased as the amount of added N increased, and PEPC activity was increased in the MN and HN treatments (Figure 4). For intermediate trees, RubisCO activity was increased in the HN treatment (Figure 4a), GS activity and GLO activity were increased in the LN, MN, and HN treatments (Figure 4b,c), and PEPC activity was increased in the MN and HN treatments (Figure 4d). For dominant trees, RubisCO activity and PEPC activity were significantly increased in the MN and HN treatments (Figure 4a,d), GS activity was increased in the HN treatment (Figure 4b), and GLO activity was increased in the LN treatment (Figure 4c).

There were no significant differences in RubisCO activity and GS activity among crown classes within each N addition treatment (Figure 4). The GLO activity differed significantly between CK and the LN treatment, and was higher in dominant trees than in suppressed and intermediate ones (Figure 4c). The PEPC activity showed significant differences between CK and the MH and HN treatments, and was higher in suppressed trees than in dominant and intermediate ones (Figure 4d).

### 2.3. Relationships Between Photosynthetic Capacity and Its Influencing Factors

Across all the experimental groups, as the N_mass_ increased, the *P*_max_, *V*_cmax,_ and *J*_max_ first increased and then decreased (Figure 5). The results of the PLSPM showed that N_mass_ positively affected enzyme activity, Chl_m_, and N:P in suppressed trees and intermediate trees. There were direct positive effects of enzyme activity and Chl_m_ on photosynthetic capacity in suppressed trees and intermediate trees (Figure 6a,b). However, N_mass_ positively affected N:P and negatively affected enzyme activity, which had a direct negative impact on the photosynthetic capacity of dominant trees (Figure 6c).

## 3. Discussion

### 3.1. Effects of N Addition on the Photosynthetic Capacity of Suppressed Trees

Nitrogen addition increases the concentration of inorganic N in the soil, which promotes N absorption by roots and leads to an increase in leaf N_mass_ [25,26,27]. Subsequently, this impacts the photosynthetic capacity. Consistent with our hypothesis, N addition significantly increased the *P*_max_, *V*_cmax_, and *J*_max_ of suppressed trees, with *P*_max_ increasing by 26%, 52%, and 56% in the LN, MN, and HN treatments, respectively, compared with CK (Figure 1 and Figure 2). This result is related to the significant increase in leaf N_mass_ after N addition. About 60% of N in the leaf is allocated to photosynthetic proteins, and more than 70% of N is located in the chloroplasts of mesophyll cells [28,29,30]. This localization of N can promote the photosynthetic capacity [31]. Our results show that as the amount of added N increased, the Chl_m_ had a similar increasing trend as *P*_max_, *V*_cmax_, and *J*_max_ in suppressed trees (Figure 3). Furthermore, the PLSPM results showed that Chl_m_ exerted the strongest influence on photosynthetic capacity (Figure 6). It should be noted that there were no differences in *P*_max_, N_mass_, and Chl_m_ between MN treatment and HN treatment (Figure 1 and Figure 3). This indicated the *P*_max_ of suppressed trees will not steadily increase as the amount of added N increased; in other words, the effect of N addition on the *P*_max_ of suppressed trees was limited. This is probably because N addition increased the N_mass_ and N:P, and *P*_max_ was limited by phosphorus. Compared to intermediate and dominant trees, the suppressed trees had lower N_mass_ and N:P, so the increased N_mass_ and N:P did not inhibit the *P*_max_ of suppressed trees.

An increase in leaf N_mass_ can affect photosynthetic enzymes, thereby influencing photosynthetic capacity [32,33]. In our study, the N addition treatments resulted in significant increases in the activities of RubisCO, GS, GLO, and PEPC in the leaves of suppressed trees (Figure 4). Increased activity of RubisCO and PEPC is beneficial for catalyzing photosynthetic carbon assimilation reactions [34,35]. The conversion of glycolic acid to glyoxylic acid is catalyzed by GLO [36,37]. Increased GLO activity can enhance the protective role of photorespiration, thereby contributing to the stability of photosynthetic capacity. Glutamine synthetase catalyzes the synthesis of glutamine from glutamate and ammonium ions (NH_4_^+^). Higher GS activity promotes the transformation of inorganic N, while the substantial synthesis of glutamine can prevent NH_4_^+^ from accumulating to toxic levels [38]. The PLSPM results showed that N addition indirectly influenced the photosynthetic capacity by directly promoting enzyme activities.

Nitrogen addition had a stronger effect on *P*_max_ than on N_mass_, resulting in an increasing trend of *PNUE* as the amount of added N increased. Notably, there were no significant differences in *P*_max_, *V*_cmax_, and *PNUE* between the MN and HN treatments. This indicated that the effect of N addition on the photosynthetic capacity of suppressed trees had a limit. That is, the photosynthetic capacity could only increase to a certain point and could not increase further with excessive N addition. Appropriate N addition can enhance the photosynthetic capacity of suppressed trees, alleviate their competitive pressure, and improve their growth and survival in the forest community.

### 3.2. Effects of N Addition on the Photosynthetic Capacity of Dominant and Intermediate Trees

Nitrogen addition significantly impacted the photosynthetic capacity of both intermediate and dominant trees. Contrary to suppressed trees, the *P*_max_ of intermediate and dominant trees initially increased and then decreased as the amount of added N increased. Especially under the HN treatment, the *P*_max_ of dominant trees was significantly lower than that of suppressed and intermediate trees. This observation aligns with our hypothesis. One reason for this result is the significant decrease in Chl_m_ of intermediate and dominant trees in the HN treatment. Another reason is that the increased N_mass_ exceeded the leaf’s tolerance and disrupted N metabolism under HN treatment [39]. Some studies have suggested that excessive N in leaves can lead to the accumulation of free amino acids, especially arginine, which is harmful to leaves [40,41]. Our results showed that *P*_max_, *V*_cmax_, and *J*_max_ had significant nonlinear relationships with N_mass_. As N_mass_ increased, *P*_max_, *V*_cmax_, and *J*_max_ first increased and then decreased, with peak values when N_mass_ was between 37.0 and 37.5 (Figure 5).

Simultaneously, excessive N may disrupt the balance of elemental stoichiometry in the leaf, thereby affecting photosynthesis [42]. The N:P is a critical indicator for measuring the balance of plant elements [43]. When the N:P is lower than 10, plants are restricted by N, and when the N:P is higher than 20, plants are restricted by P [44,45]. Our results show that N addition had no significant impact on P_mass_, but it significantly increased the N_mass_ and N:P. Especially in the HN treatment, the N:P of intermediate and dominant trees was 21.27 and 21.74, respectively (Figure 3). The disruption of the balance between N and P in the leaf may be one of the reasons for the decrease in photosynthetic capacity [46,47]. The PLSPM results showed that N addition indirectly limited the photosynthetic capacity by promoting N:P. This indicates that intermediate trees and dominant trees may be restricted by P in the HN treatment.

Changes in the activity of photosynthetic enzymes also contributed to the decrease in photosynthetic capacity in the HN treatment. Our results show that the activities of RubisCO, PEPC, and GS were highest in the HN treatment for all crown classes; the activities of RubisCO and GS did not differ significantly among the crown classes; and the activity of PEPC was lower in dominant trees than in suppressed and intermediate ones (Figure 4). However, in the HN treatment, the N_mass_ was significantly higher in intermediate and dominant trees than in suppressed trees. This indicates that the increase of N_mass_ enhanced the activity of photosynthetic enzymes, but when N_mass_ reached a certain value, this effect was no longer significant or even inhibitory. Moreover, both the GLO activity and *P*_max_ of intermediate trees and dominant trees significantly decreased in the HN treatment. The decrease in GLO activity could potentially attenuate photorespiration, consequently lowering the efficiency of photosystem II, and ultimately resulting in decreased photosynthetic capacity [48]. Nitrogen addition can also affect the photosynthetic capacity by affecting the amount of enzymes. In this study, we determined the activity of photosynthetic enzymes, but not their contents. We intend to further explore the effects of N addition on the contents and activities of enzymes in further research.

### 3.3. Differences in Photosynthetic Capacity Among Crown Classes

In CK, the photosynthetic parameters (e.g., *P*_max_, *V*_cmax_, and *J*_max_) of suppressed trees were significantly lower than those of intermediate trees and dominant trees (Figure 1). Suppressed trees are in the lower forest layer and have a weaker ability to compete for soil nutrients. This will result in low leaf N_mass_ and photosynthetic capacity [49]. Moreover, in CK, there were no significant differences in *P*_max_, *V*_cmax_, and *J*_max_ between intermediate trees and dominant trees. This might be because the intermediate trees had a similar ability to obtain soil nutrients as dominant trees because the differences in N_mass_ and P_mass_ were not significantly different between these two crown classes. In addition, dominant trees may be subject to light stress, which would limit their photosynthetic capacity. This may also explain why GLO activity was significantly higher in dominant trees than in intermediate and suppressed ones (Figure 4).

Nitrogen addition had a significant impact on the photosynthetic capacity of trees across all crown classes, but the degree of impact varied. In the LN treatment, the *P*_max_ and *V*_cmax_ of each crown class showed similar increases, and suppressed trees showed the largest increase in *J*_max_ (Figure 2). This suggested that the LN treatment strongly affected the regeneration rate of ribulose-1,5-bisphosphate in suppressed trees. Compared with CK, the LN treatment did affect the magnitude of differences in *P*_max_, *V*_cmax_, and *J*_max_ among the crown classes. In the MN treatment, the *P*_max_, *V*_cmax_, and *J*_max_ of intermediate trees were significantly greater than those of suppressed trees and dominant trees. In the HN treatment, the *P*_max_, *V*_cmax_, and *J*_max_ were lower in the dominant trees than in the suppressed and intermediate trees. Overall, N addition altered the magnitude of differences in photosynthetic capacity among crown classes. The PLSPM results also showed that there were differences in the direct pathways by which N affected photosynthetic capacity among the three crown classes. This may be related to the N saturation threshold of the trees in the different crown classes. In CK, the leaf N_mass_ of suppressed trees did not reach its saturation threshold [10,11,12]. As the amount of added N increased, the leaf N_mass_ of suppressed trees increased but did not exceed the saturation threshold, allowing the leaves to allocate more N to photosynthetic components [50,51]. In CK, the N_mass_ in the leaves of intermediate trees and dominant trees were close to their saturation threshold, as the amount of added N increased, the N_mass_ continued to increase and exceeded the threshold. This may have disrupted the balance of photosynthetic metabolism and inhibited the photosynthetic capacity of intermediate trees and dominant trees [52,53]. As we have discussed above, N addition increased the N:P and led to insufficient phosphorus supply. This will disrupt energy metabolism and induce oxidative stress, leading to reactive oxygen species accumulation, which compromises the stability of photosystem II.

These findings deepen our understanding of how N addition affects the photosynthetic capacity of *L*. *gmelinii*, which will be useful for clarifying the effect of atmospheric N deposition on boreal forests. Suppressed trees grow slowly and have a low survival rate in the forest, but N deposition can increase their photosynthetic capacity, which is beneficial for their growth and survival. This helps to maintain the stability and enhance the carbon sequestration of forest ecosystems. Appropriate N deposition can also increase the photosynthetic capacity of intermediate trees and dominant trees. However, excessive N deposition can reduce the increase in photosynthetic capacity of intermediate trees and dominant trees, and even inhibit the photosynthetic capacity of dominant trees [54]. Our findings clarify the effect of N addition on photosynthetic capacity was different among trees in different crown classes, and the crown class should be considered in the N deposition model. Meanwhile, when studying the impact of N deposition on *L*. *gmelinii*, attention should be paid to crown-classed trees instead of standard trees.

## 4. Materials and Methods

### 4.1. Site Description

The study site was situated within the Nanwenghe National Nature Reserve in the Greater Khingan Mountains (51°05′−51°39′ N, 125°07′−125°50′ E). This region features a cold temperate continental monsoon climate, with an average annual temperature of −2.7 °C, a temperature low of −48 °C, and a temperature high of 36 °C. The average annual precipitation during the study period was 500 mm, with an accumulated temperature of about 1500 °C for temperatures ≥ 10 °C, and a frost-free period of 90–100 days. The soil type was brown coniferous forest soil.

The vegetation at the study site was a natural secondary forest that had regenerated after logging. The dominant tree species was *L. gmelinii*, and the associated tree species was *Betula platyphylla*. The main understory herbaceous plants included *Pyrola asarifolia*, *Maianthemum bifolium*, *Eguisetum sylvalicum*, and *Carex tristachya*. During sampling in 2023, the growth density at the site was 2852 ± 99 trees·ha^−1^, the average diameter at breast height (DBH) was 9.38 ±0.99 cm, the maximum tree age was 42 years, and the canopy density was 0.76 ± 0.30.

### 4.2. Experimental Design

The experiment was established with a randomized complete block design, consisting of three blocks. Each block contained four 20 m × 20 m plots with a 10 m buffer in between. Four levels of N addition were established in the four plots: control (CK, 0 Kg N·ha^−1^·yr^−1^), low N (LN, 25 Kg N·ha^−1^·yr^−1^), medium N (MN, 50 Kg N·ha^−1^·yr^−1^), and high N (HN, 75 Kg N·ha^−1^·yr^−1^). The N addition levels were chosen to simulate future N deposition of 1–3 times the current N deposition level in the study region (25 kg N·ha^−1^·yr^−1^) [24]. The N addition experiment was started in May 2011. In each plot, N addition was carried out five times from early May to early September each year. For each N addition level, the corresponding amount of NH_4_NO_3_ was dissolved in 32 L water and evenly sprayed on the forest floor using a sprayer. To eliminate the effect of differences in water supply, the control plots were sprayed with the same amount of pure water.

### 4.3. Measurements of Leaf Gas Exchange

The leaf gas exchange measurements were conducted in late July 2023. First, the DBH of all trees in each plot was measured, and the trees were categorized into three crown classes: suppressed trees (3 cm < DBH ≤ 6 cm), intermediate trees (6 cm < DBH ≤ 16 cm), and dominant trees (16 cm < DBH ≤ 25 cm). In each plot, three sample trees were selected from each crown class. One wood core (5.15 mm in diameter) per sample tree was collected from the trunk at breast height to confirm the sample trees were of a similar age. In each plot, three standard branches were cut from the middle canopy of each sample tree, and the needles from the 2–3 cm portion at the end of three short shoots of each standard branch were used for light response curves and CO_2_ response curves measurement with a portable infrared gas analyzer (LI-6800, Li-Cor Inc., Lincoln, NE, USA). The maximum net photosynthetic rate (*P*_max_), maximum carboxylation rate (*V*_cmax_), and maximum electron transport rate (*J*_max_) were calculated using empirical equations [55].

### 4.4. Measurements of Leaf Traits

The needles sampled for gas exchange measurements were harvested and placed in a cooler at 4 °C. In the laboratory, the needles were scanned, dried, and then weighed to calculate the specific leaf area (SLA = leaf area/leaf dry mass).

About 100 g needles on the short shoot of each branch were sampled, randomly divided into three portions, and placed in liquid N. In the laboratory, one portion was used to measure the activity of ribulose-1,5-bisphosphate carboxylase (RubisCO), glutamine synthetase (GS), glycolate oxidase (GLO), and phosphoenolpyruvate carboxylase (PEPC) using the RubisCO kit, the GS kit, the GLO kit, and the PEPC kit (Keming Biotechnology Inc., Suzhou, China), respectively. The absorbance of the reaction mixtures was measured using a spectrophotometer (Evolution 300, Thermo Fisher Scientific Inc., Waltham, MA, USA). Another portion was dried at 65 °C to a constant weight (accuracy 0.0001 g). The dried samples were then ground, sieved, and pressed into a tablet for analyses of N content per unit mass (N_mass_) and phosphorus content per unit mass (P_mass_) using a laser spectral elemental analysis system (J200 Tandem LA-LIBS Instrument, Applied Spectra, Fremont, CA, USA). The N:P was calculated by dividing the N_mass_ by P_mass_. The leaf N concentration per unit area (N_area_) was obtained by dividing the N_mass_ by SLA. The photosynthetic N-use efficiency (*PNUE*) was obtained by dividing *P*_max_ by N_area_ [56]. The third portion was used to determine the contents of photosynthetic pigments [57]. The needles were ground, extracted with a 1:1 mixture of acetone and ethanol, and then the absorbance of the solvent was determined at 645 nm and 663 nm using a spectrophotometer (Evolution 300, Thermo Fisher).

### 4.5. Data Analyses

All statistical analyses were conducted using R software (version 4.4.2, The R Foundation for Statistical Computing, Vienna, Austria). Data were subjected to analysis of variance (ANOVA) to test the main effects (N addition treatment, crown class) on photosynthetic characteristics and related needle traits with the “agricolae 1.3-7” package. The data were tested for normal distribution and homogeneity of variance before ANOVA. Multiple comparisons of photosynthetic characteristics and related needle traits were conducted with the “multcomp 1.4-26” package. A regression model was applied to explore potential relationships between photosynthetic characteristics and needle traits with the “car 3.1-2” package. Partial least squares path models (PLSPMs) were applied to analyze the driving factors of photosynthetic capacity with the “lavaan 0.6-19” package.

For each crown class, the difference in photosynthetic characteristics in each N addition treatment compared with the control was calculated as follows:Δ*X*_i_ = (*X*_i-addition_ − *X*_i-control_)/*X*_i-control_ × 100% (1)
where *X*_addition_ and *X*_control_ represent each variable (e.g., *P*_max_, *V*_cmax_, *J*_max_, and *PNUE*) measured in the N addition treatment and control (CK), respectively, and i is the crown class.

## 5. Conclusions

Nitrogen addition significantly affected the photosynthetic capacity of *L*. *gmelinii*, and the effect varied with the amount of added N and the crown class. Nitrogen addition resulted in significant increases in the photosynthetic capacity of suppressed trees and intermediate trees. As the amount of added N increased, the *P*_max_ increased for suppressed trees and first increased and then decreased for intermediate trees. For dominant trees, the *P*_max_ significantly increased in the LN and MN treatments and significantly decreased in the HN treatment. Overall, the appropriate amount of N addition was beneficial for the photosynthetic capacity of *L*. *gmelinii*. The changes in leaf N_mass_, Chl_m_, and photosynthetic enzyme activities under N addition contributed to the variations in photosynthetic capacity among trees in different crown classes. The photosynthetic capacity of suppressed trees and intermediate trees benefitted most from N addition, and this altered the magnitude of differences in photosynthetic capacity among the crown classes. For example, the suppressed trees had the minimum *P*_max_ in CK, and the dominant trees had the minimum *P*_max_ in the HN treatment. Therefore, the crown class should be considered when studying the effect of N deposition on boreal forests. These results can help us develop more rational forest management strategies and accurately predict the effect of nitrogen deposition on forest structure and function.

## Figures and Tables

**Figure 1 plants-14-01056-f001:**
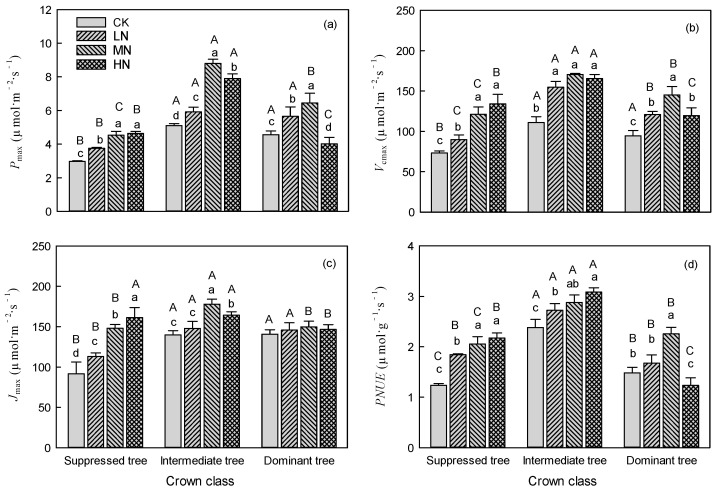
Effects of N addition on (**a**) maximum net photosynthetic rate (*P*_max_), (**b**) maximum carboxylation rate (*V*_cmax_), (**c**) maximum electron transfer rate (*J*_max_), and (**d**) photosynthetic nitrogen use efficiency (*PNUE*) of trees in different crown classes. Error bars represent standard errors (*n* = 3), and different capital letters indicate significant differences among crown classes in the same N addition treatment (*α* = 0.05), and different lowercase letters indicate significant differences among N addition treatments within the same crown class (*α* = 0.05). CK: control, LN: low N addition treatment, MN: middle N addition treatment, HN: high N addition treatment.

**Figure 2 plants-14-01056-f002:**
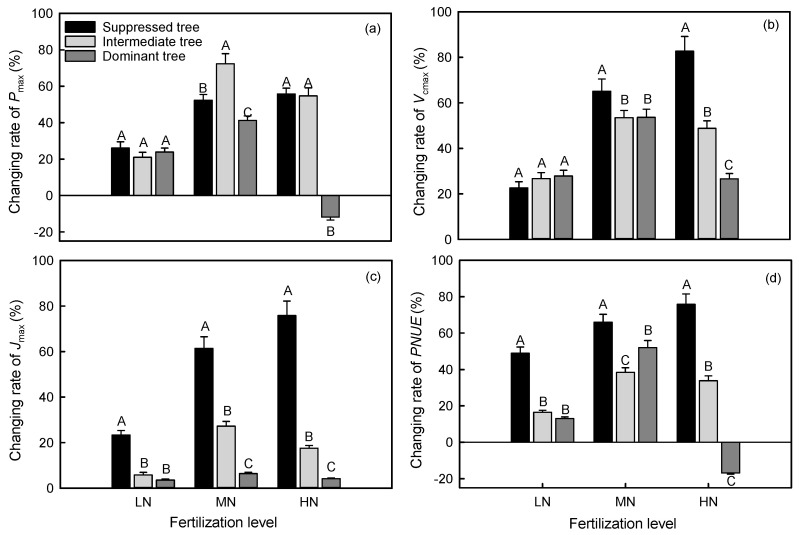
Differences in (**a**) maximum net photosynthetic rate (*P*_max_), (**b**) maximum carboxylation rate (*V*_cmax_), (**c**) maximum electron transfer rate (*J*_max_), and (**d**) photosynthetic nitrogen use efficiency (*PNUE*) among trees in different crown classes in the same N addition treatments. Error bars represent standard errors (*n* = 3), and different capital letters indicate significant differences among crown classes in the same N addition treatment (*α* = 0.05). LN: low N addition treatment, MN: middle N addition treatment, HN: high N addition treatment.

**Figure 3 plants-14-01056-f003:**
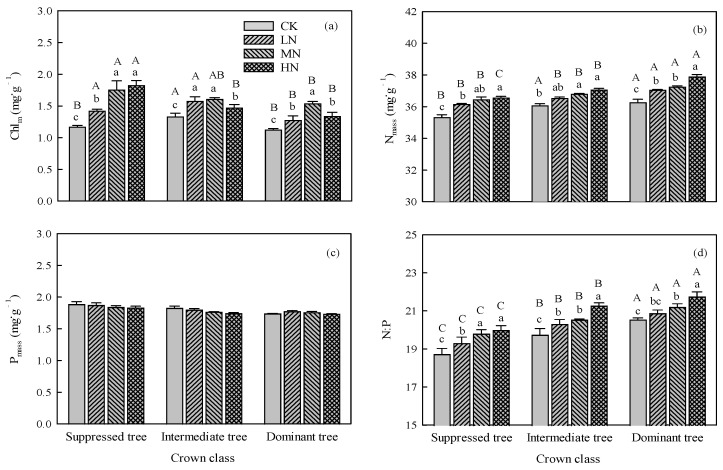
Effects of N addition on (**a**) chlorophyll content (Chl_m_), (**b**) nitrogen content (N_mass_), (**c**) phosphorous content (P_mass_), and (**d**) ratio of nitrogen to phosphorous (N:P) in trees in different crown classes. Error bars represent standard errors (*n* = 3), and different capital letters indicate significant differences among crown classes under the same N addition treatment (*α* = 0.05), and different lowercase letters indicate significant differences among N addition treatments within the same crown class (*α* = 0.05). CK: control, LN: low N addition treatment, MN: middle N addition treatment, HN: high N addition treatment.

**Figure 4 plants-14-01056-f004:**
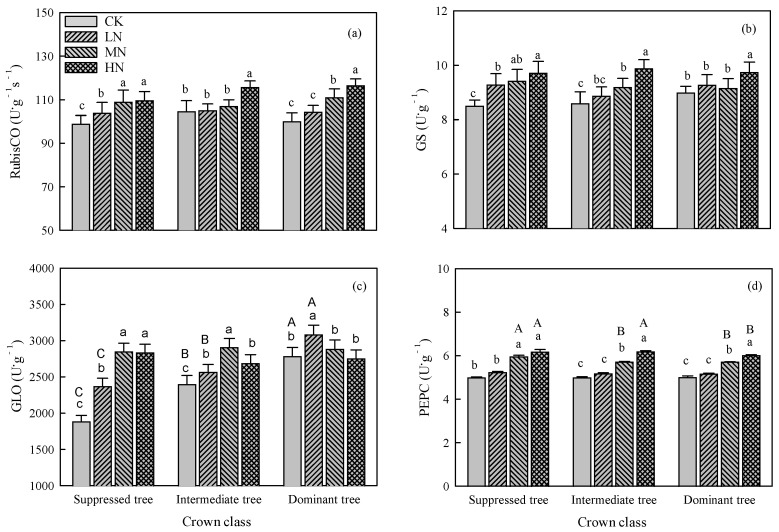
Effects of N addition on activity of (**a**) 1, 5-ribulose bisphosphate carboxylase (RubisCO), (**b**) glutamine synthetase (GS), (**c**) glycolate oxidase (GLO), and (**d**) phosphoenolpyruvate carboxylase (PEPC) in trees in crown classes. Error bars represent standard errors (*n* = 3), and different capital letters indicate significant differences among crown classes in the same N addition treatment (*α* = 0.05), and different lowercase letters indicate significant differences among N addition treatments within the same crown class (*α* = 0.05). CK: control, LN: low N addition treatment, MN: middle N addition treatment, HN: high N addition treatment.

**Figure 5 plants-14-01056-f005:**
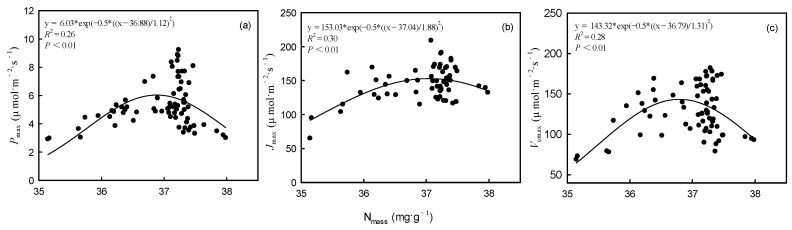
Relationships between nitrogen content (N_mass_) and maximum net photosynthetic rate (*P*_max_) (**a**), maximum electron transport rate (*J*_max_) (**b**), and maximum carboxylation rate (*V*_cmax_) (**c**).

**Figure 6 plants-14-01056-f006:**
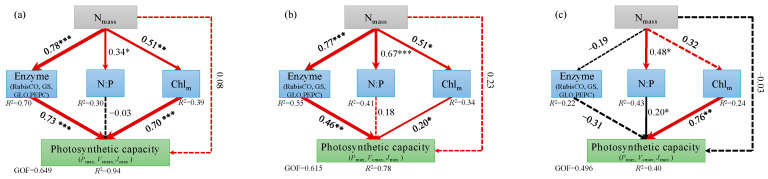
Partial least squares path modeling to demonstrate major pathways of the influences of N_mass_, photosynthetic enzyme activities, nitrogen–phosphorus ratio (N:P), and chlorophyll (Chl_m_) on the photosynthetic capacity of *L. gmelinii* for (**a**) suppressed trees, (**b**) intermediate trees, and (**c**) dominant trees. Solid and dashed arrows represent significant effect and no significant effect, respectively. Red and black arrows indicate positive and negative effects, respectively. Line thickness represents the strength of the causal relationship. Values near arrows indicate standardized coefficients for each causal path and covariance (*, *p*  <  0.05; **, *p  *<  0.01; ***, *p*  <  0.001). Values of *R*^2^ represent the amount of variation explained in response variables. GOF indicates the goodness of fit of the entire model.

**Table 1 plants-14-01056-t001:** ANOVA results for larch leaf traits *.

Variable	Crown Class		Nitrogen Addition		Crown Class × Nitrogen Addition	
	*F*	*p*	*F*	*p*	*F*	*p*
*P* _max_	50.85	<0.01	17.37	<0.01	6.67	<0.01
*V* _cmax_	32.63	<0.01	14.93	<0.01	3.26	<0.01
*J* _max_	8.96	<0.01	9.14	<0.01	2.48	0.04
N_mass_	8.31	<0.01	17.79	<0.01	3.92	<0.01
P_mass_	17.44	<0.01	1.64	0.19	0.79	0.58
N:P	28.00	<0.01	6.78	<0.01	2.32	0.05
*PNUE*	56.84	<0.01	9.17	<0.01	4.12	<0.01
Chl_m_	11.54	<0.01	15.97	<0.01	3.37	<0.01
SLA	47.50	<0.01	1.84	0.15	6.18	<0.01
RubisCO	6.38	<0.01	59.42	<0.01	2.69	0.02
GS	1.72	0.13	41.97	<0.01	3.99	0.02
GLO	5.29	0.10	1.72	0.18	2.30	0.05
PEPC	7.95	<0.01	199.04	<0.01	3.41	<0.01

* *P*_max_: maximum net photosynthetic rate; *V*_cmax_: maximum carboxylation rate; *J*_max_: maximum electron transfer rate; N_mass_: mass-based nitrogen content; P_mass_: mass-based phosphorus content; N:P: ratio of nitrogen to phosphorous; *PNUE*: photosynthetic nitrogen-use efficiency; Chl_m_: chlorophyll content; SLA: specific leaf area; RubisCO: 1, 5-ribulose bisphosphate carboxylase; GS: glutamine synthetase; GLO: glycolate oxidase; PEPC: phosphoenolpyruvate carboxylase. *F*: *F*-ratio; *p*: *p*-value. The same below.

## Data Availability

The data presented in this study are available on request from the corresponding author.

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
