# Peer review of "Response of Photosynthetic Capacity to Nitrogen Addition in Larix gmelinii Trees in Different Crown Classes"

_plants, 2025, doi:10.3390/plants14071056_

Round 1
Reviewer 1 Report
Comments and Suggestions for Authors
After reviewing the paper entitled Response of Photosynthetic Capacity to Nitrogen Addition in Larix gmelinii Trees in Different Crown Classes, I consider that it is well written, the material and methods are in accordance with the objectives of the work, and the authors have made a discussion and conclusions that fit the results obtained.
It is also true that there are a large number of publications on the species under study, and many papers on the effect of nitrogen levels on this species. Nevertheless, I believe that the approach is novel.
Next I make some comments for the authors
Line 76 and 77. The addition of nitrogen did not increase the photosynthetic parameters in suppressed and intermediate trees. If we look at figure 1, there is an increase in photosynthetic parameters up to MN. Above these amounts of nitrogen, there is either no effect or even a negative effect of nitrogen with a decrease.
On the other hand, if it is indicated on line 79 for the intermediated trees
Line 95-97. Not all parameters in intermediated and dominant trees strongly increased their values; for example, Vcmax and PNUE in intermeadiate three did not increase in MN. Neither did Jmax in dominat three in MN.
In general, the authors should correct these errors in the text that do not correspond to what is shown in the figures.
Line 125-127. The same is observed in section 2.2. The factors related to photosynthesis are not always increased. It would be important to emphasise the fact that in Chlm, in intermediate and dominate three, the levels decrease in HN.
Line 130-131. This statement is not correct. In Chlm and N:P there were no significant differences between crown classes for CK. Along the same line, the authors refer to figure 3a, but should refer to figures 3b and 3d.
In general, this paragraph should also be revised and the text should be rewritten to make it consistent with the figures and to match the significant differences mentioned in the text with those observed in the figures.
Line 195: in the same sense as the results, the increase in N did not imply an increase in HN. It would be good to explain what happens at these N levels, apart from commenting that it does not imply an increase in photosynthesis. Nor do the authors explain what happens with Pmass and N:P.
Minor comments:
The authors should review the bibliography.
They should indicate the abbreviated journal name in citations 13, 14, 20, and 37 and 39.
Reference 18: begin the name of the journal “science” with a capital letter.
From line 542 onwards, the font size appears greater than in the rest of the text.
Line 551: indicates the scientific name in italics.
Author Response
For research article
|
Response to Reviewer 1 Comments |
||
|
1. Summary |
|
|
|
Thank you very much for taking the time to review this manuscript. Please find the detailed responses below and the corresponding revisions in track changes in the re-submitted files.
|
||
|
2. Questions for General Evaluation |
Reviewer’s Evaluation |
Response and Revisions |
|
Does the introduction provide sufficient background and include all relevant references? |
Yes |
Thanks |
|
Is the research design appropriate? |
Yes |
Thanks |
|
Are the methods adequately described? |
Yes |
Thanks |
|
Are the results clearly presented? |
Yes |
Thanks |
|
Are the conclusions supported by the results?
|
Yes |
Thanks |
|
3. Point-by-point response to Comments and Suggestions for Authors |
||
|
Comments 1: After reviewing the paper entitled Response of Photosynthetic Capacity to Nitrogen Addition in Larix gmelinii Trees in Different Crown Classes, I consider that it is well written, the material and methods are in accordance with the objectives of the work, and the authors have made a discussion and conclusions that fit the results obtained. It is also true that there are a large number of publications on the species under study, and many papers on the effect of nitrogen levels on this species. Nevertheless, I believe that the approach is novel. |
||
|
Response 1: Thanks for your comments. |
||
|
Comments 2: Line 76 and 77. The addition of nitrogen did not increase the photosynthetic parameters in suppressed and intermediate trees. If we look at figure 1, there is an increase in photosynthetic parameters up to MN. Above these amounts of nitrogen, there is either no effect or even a negative effect of nitrogen with a decrease. On the other hand, if it is indicated on line 79 for the intermediated trees. |
||
|
Response 2: Thanks for your comments. The result we want to express is that the low nitrogen, medium nitrogen, and high nitrogen treatments are significantly greater than the control treatment. The description of multiple comparisons for figure 1 is not accurate. We have revised these sentences. Comments 3: Line 95-97. Not all parameters in intermediated and dominant trees strongly increased their values; for example, Vcmax and PNUE in intermeadiate three did not increase in MN. Neither did Jmax in dominat three in MN. In general, the authors should correct these errors in the text that do not correspond to what is shown in the figures. Response 3: Thanks for your comments. Revised. Comments 4: Line 125-127. The same is observed in section 2.2. The factors related to photosynthesis are not always increased. It would be important to emphasise the fact that in Chlm, in intermediate and dominate three, the levels decrease in HN. Response 4: Thanks for your comments. Revised. Comments 5: Line 130-131. This statement is not correct. In Chlm and N:P there were no significant differences between crown classes for CK. Along the same line, the authors refer to figure 3a, but should refer to figures 3b and 3d. In general, this paragraph should also be revised and the text should be rewritten to make it consistent with the figures and to match the significant differences mentioned in the text with those observed in the figures. Response 5: Thanks for your comments. Revised. Comments 6: Line 195: in the same sense as the results, the increase in N did not imply an increase in HN. It would be good to explain what happens at these N levels, apart from commenting that it does not imply an increase in photosynthesis. Nor do the authors explain what happens with Pmass and N:P. Response 6: Thanks for your comments. Revised. Comments 7: Minor comments: The authors should review the bibliography. They should indicate the abbreviated journal name in citations 13, 14, 20, and 37 and 39. Reference 18: begin the name of the journal “science” with a capital letter. From line 542 onwards, the font size appears greater than in the rest of the text. Line 551: indicates the scientific name in italics. Response 7: Thanks for your comments. Revised. |
||
|
4. Response to Comments on the Quality of English Language |
||
|
Point 1: The English is fine and does not require any improvement. |
||
|
Response 1: Thanks. |
||

Reviewer 2 Report
Comments and Suggestions for Authors
Review on “Response of Photosynthetic Capacity to Nitrogen Addition in Larix gmelinii Trees in Different Crown Classes”
The authors examined the effects of different nitrogen treatments (0, 25, 50, and 75 kg nitrogen ha/year) on Larix gmelinii tress with different crown classes. The authors measured leaf gas exchange and leaf specifications. They stated that the photosynthetic capability of Larix gmelinii was strongly impacted by nitrogen addition, with the effect varying depending on the crown class and the amount of N applied. However, based on physiological studies, efficiency of photosynthesis depends on leaf position, leaf angle, and number of leaves, etc. - not just in Larix gmelinii but plants more generally. In addition, nitrogen has impact on the efficiency of photosynthesis is well-established too. Therefore, I do not see any new findings presented in this statement.
The manuscript is well-written. The Introduction section provides a clear background and presents the goal of the study. The research data is well-presented, and the findings are statistically analyzed.
Author Response
For research article
|
Response to Reviewer 2 Comments |
||
|
1. Summary |
|
|
|
Thank you very much for taking the time to review this manuscript. Please find the detailed responses below and the corresponding revisions in track changes in the re-submitted files.
|
||
|
2. Questions for General Evaluation |
Reviewer’s Evaluation |
Response and Revisions |
|
Does the introduction provide sufficient background and include all relevant references? |
Yes |
Thanks |
|
Are all the cited references relevant to the research? |
Yes |
Thanks |
|
Is the research design appropriate? |
Yes |
Thanks |
|
Are the methods adequately described? |
Yes |
Thanks. |
|
Are the results clearly presented? |
Yes |
Thanks. |
|
Are the conclusions supported by the results? |
Must be improved |
Thanks. The conclusions had been revised. |
|
3. Point-by-point response to Comments and Suggestions for Authors |
||
|
Comments 1: The authors examined the effects of different nitrogen treatments (0, 25, 50, and 75 kg nitrogen ha/year) on Larix gmelinii tress with different crown classes. The authors measured leaf gas exchange and leaf specifications. They stated that the photosynthetic capability of Larix gmelinii was strongly impacted by nitrogen addition, with the effect varying depending on the crown class and the amount of N applied. However, based on physiological studies, efficiency of photosynthesis depends on leaf position, leaf angle, and number of leaves, etc. - not just in Larix gmelinii but plants more generally. In addition, nitrogen has impact on the efficiency of photosynthesis is well-established too. Therefore, I do not see any new findings presented in this statement. |
||
|
Response 1: Thanks for your comments. I agree with your viewpoint that a good manuscript should have an interesting finding or solve a scientific question. I also strive to provide well-researched, informative, and positive manuscript to readers. After reviewing the literature, I found that most nitrogen addition studies mainly focused on standard trees and ignored differences among trees of the same species, especially in natural forests. So, we studied the response of trees in different crown classes to nitrogen addition. We found nitrogen addition had significant effect on photosynthetic capacity of Larix gmelinii, and the effect varied with the amount of added nitrogen and the crown class. The photosynthetic capacity of suppressed trees and intermediate trees benefitted most from N addition, and this altered the magnitude of differences in photosynthetic capacity among the crown classes. These findings can help us develop more rational forest management strategies and accurately predict the effect of nitrogen deposition on forest structure and function. |
||
|
Comments 2: The manuscript is well-written. The Introduction section provides a clear background and presents the goal of the study. The research data is well-presented, and the findings are statistically analyzed. Response 2: Thanks. 4. Response to Comments on the Quality of English Language. Point 1: The English is fine and does not require any improvement. Response 1: Thanks. |
||

Reviewer 3 Report
Comments and Suggestions for Authors
The present reseach evaluates the impact of N fertilization on the growth, N absorption and photosynthetic activity in Larix gmelinii Trees in Different Crown Classes.
The abstract is clear and presents the essential results and conclusions of the work.
The introduction is also complete clear with good and pertinent references (except one which should be remplaced, please the document attached). The research objectives are clearly stated.
The results are really well described, the tables and graphic are clear with good statistical analysis. The legened are really complete. The discussion is complete with good and also pertinent references.
The material and method section is also clear, only one minor point should clarified (see attached).
The conclusions correspond to the results obtained, maybe there too many results present and less real conclusions than expected.

Author Response
For research article
|
Response to Reviewer 3 Comments |
|||||
|
1. Summary |
|
|
|||
|
Thank you very much for taking the time to review this manuscript. Please find the detailed responses below and the corresponding revisions in track changes in the re-submitted files. |
|||||
|
2. Questions for General Evaluation |
Reviewer’s Evaluation |
Response and Revisions |
|||
|
Does the introduction provide sufficient background and include all relevant references? |
Yes |
Thanks |
|||
|
Are all the cited references relevant to the research? |
Yes |
Thanks |
|||
|
Is the research design appropriate? |
Yes |
Thanks |
|||
|
Are the methods adequately described? |
Can be improved |
Thanks for your suggestion. More detailed information of the method was given. |
|||
|
Are the results clearly presented? |
Yes |
Thanks |
|||
|
Are the conclusions supported by the results? |
Yes |
Thanks |
|||
|
3. Point-by-point response to Comments and Suggestions for Authors |
|||||
|
Comments 1: The present research evaluates the impact of N fertilization on the growth, N absorption and photosynthetic activity in Larix gmelinii Trees in Different Crown Classes. The abstract is clear and presents the essential results and conclusions of the work. The introduction is also complete clear with good and pertinent references (except one which should be replaced, please the document attached). The research objectives are clearly stated. The results are really well described, the tables and graphic are clear with good statistical analysis. The legend is really complete. The discussion is complete with good and also pertinent references. The material and method section is also clear, only one minor point should clarified (see attached). The conclusions correspond to the results obtained, maybe there too many results present and less real conclusions than expected. Response 1: Thanks for your comments. |
|||||
|
Comments 2: Reference{17] is not really suitable to support this reference about ecological distribution as the main treated topic is about N. Response 2: Thanks for your suggestion. I have replaced it with an appropriate reference. |
|||||
|
Comments 3: ”……was tree” Inverse the order. Response 3: Thanks. Revised. |
|||||
|
Comments 4: Indicate the meaning of F-value and P-value in the legend. Response 4: Revised. |
|||||
|
Comments 5: How was it collected at which heigh was placed the analyzer, how many measurements were made in each plot? Response 5: Revised.
|
|||||

Round 2
Reviewer 2 Report
Comments and Suggestions for Authors
-